# Long-Term Effects of Low-Level Blast Exposure and High-Caliber Weapons Use in Military Special Operators

**DOI:** 10.3390/brainsci12050679

**Published:** 2022-05-23

**Authors:** Melissa Hunfalvay, Nicholas P. Murray, William T. Creel, Frederick R. Carrick

**Affiliations:** 1RightEye LLC, 7979 Old Georgetown Rd, Suite 801, Bethesda, MD 20814, USA; 2Department of Kinesiology, East Carolina University, Minges Coliseum 166, Greensville, NC 27858, USA; murrayni@ecu.edu; 3Neurology Department, Adler University, 17 N Dearborn St, Chicago, IL 60602, USA; wcreel@adler.edu; 4Neurology Department, College of Medicine, University of Central Florida, Orlando, FL 32827, USA; 5Centre for Mental Health Research in Association with University of Cambridge, Cambridge CB2 1TN, UK; 6Department of Health Professions Education, MGH Institute for Health Professions, Boston, MA 02129, USA

**Keywords:** SPEM, smooth-pursuit eye movements, oculomotor, TBI, military

## Abstract

Chronic low-level blast exposure has been linked with neurological alterations and traumatic brain injury (TBI) biomarkers. Impaired smooth-pursuit eye movements (SPEM) are often associated with TBI. The purpose of this study was to determine whether long-term operators of low-level blast exposure or high-caliber weapons use displayed oculomotor behaviors that differed from controls. Twenty-six members of an elite military unit performed a computerized oculomotor testing task using an eye tracker and completed a concussion assessment questionnaire. The participants were split into a blast exposure group and control group. The blast exposure group had a history of exposure to low-level blasts or high-caliber weapon use. The results revealed significant differences in SPEM, saccades, and fixations between the blast exposure group and control group. The blast exposure group’s eye movements were slower, stopped at more frequent points when following a target, traveled further from the target in terms of both speed and direction, and showed higher rates of variation and inefficiency. Poor oculomotor behavior correlated with a higher symptom severity on the concussion assessment questionnaire. Military special operators exposed to long-term low-level blasts or high-caliber weapons usage displayed an impaired oculomotor behavior in comparison to controls. These findings further our understanding of the impact of long-term low-level blast exposure on the oculomotor behavior of military special operators and may inform practical implications for military training.

## 1. Introduction

There has been a growing interest in understanding the effects of long-term low-level blast exposure and high-caliber weapon use among military personnel due to concern over potential adverse health outcomes [1]. Recent findings suggest that chronic exposure to low-level blasts may be implicated in neurological alterations and elevated biomarkers associated with traumatic brain injury (TBI) [2,3].

TBI is a unique and prominent cause of morbidity in military life [4,5]. Given the prevalent nature of TBI in military populations, there is an evolving need to understand how repeated low-level blast exposure and high-caliber weapon use may present an additional threat to service members’ mental health and well-being. Determining the impact of low-level blast exposure on cognitive functioning has proven especially challenging given the evasive nature of concussive injury that is invisible to the eye. However, eye-tracking technology offers a quantitative, non-invasive, and sensitive solution that can provide a detailed insight into the brain and cognitive functioning of military personnel following low-level blast exposure [6].

Eye-tracking technology is a uniquely equipped tool for accurately and efficiently measuring objective TBI biomarkers, including for differentiating the severity of TBI [7,8,9,10,11]. Eye tracking has been used to assess neurological functioning, oculomotor assessment, neurocircuitry abnormalities, and even map oculomotor dysfunction to associated brain regions [12,13,14].

Oculomotor assessments can be subdivided into discrete eye movements including smooth pursuits, fixations, and saccades [15]. Smooth pursuits use predictive eye-tracking movements to stabilize a moving target on the fovea, fixations involve maintaining a fixed eye contact on a visual target input, and saccades are short and quick eye movements between two points. Collectively, smooth pursuits, fixations, and saccades can reveal neurological abnormalities through a comprehensive smooth pursuit eye movement (SPEM) assessment.

The smooth pursuit system is very complex, and while not yet fully understood, it is what allows humans to predictively track moving objects [16,17,18]. After visual inputs are processed in the striate cortex, information is relayed to specialized neurons in the extrastriate areas, which extend to the brainstem and allow communication with the cerebellum [17]. The cerebellum is intimately involved in visual perception and is critical in generating pursuits [19]. The pursuits themselves are primarily controlled by the frontal eye field and deeper brain structures including the basal ganglia and superior colliculus [17].

Circular smooth pursuits (CSP) have been observed to activate the visual cortex bilaterally and the right inferior parietal sulcus [20]. Neuron firing has also been detected in the medial temporal lobes and premotor cortex with a marked depressed activity in the insula and anterior cingulate [21]. Evidently, the complexity and breadth of the smooth pursuit networks makes the system susceptible to damage from TBI.

CSP can be tracked to measure fixation percentages. Concussed individuals have higher fixation percentages as they are constantly falling behind the target, requiring their eyes to saccade to catch up to the target [22]. Fixation is a conscious process involving a network of brain regions including the parietal eye field, supplemental eye field, V5 and V5A areas, and the dorsolateral prefrontal cortex [17]. The inability to smoothly track moving objects causes abnormal fixations that may be indicative of damage to these networks [9].

Variance is another metric that is obtained from tracking CSP. The neurons of the pons contribute to variance measures, as they are tuned to the eye velocity and can be stimulated to change the velocity of pursuits. The pontine nuclei project to the cerebellum, which is involved in the online correction of velocity during pursuit. A high smooth pursuit variance shows inefficiencies in eye movement that are often indicative of TBI [23].

The saccadic system includes several brain structures including the brain stem, pons, midbrain, and cerebral cortex [17]. Saccades are generated by burst neuron circuits in the brain stem, which activate motor signals that control the extraocular muscles in the eye [17]. Multiple studies have shown that saccadic impairment is associated with TBI [10,24].

More recent approaches to understanding the impact of low-level blast exposure on brain health have revealed elevated biomarkers associated with traumatic brain injury and brain diseases in military populations [3]. These results further validate efforts towards identifying objective biomarkers that can inform clinically relevant diagnostic tools.

In contrast, most current methods of TBI diagnosis involve an element of subjectivity that lacks sensitivity to intricate symptomology [7]. The Sport Concussion Assessment Tool (SCAT) is among the most respected concussion inventories and is widely used by team physicians [25]. The assessment has demonstrated sensitivity and specificity measures of 96% and 81%, respectively, suggesting a strong validity and reliability [26]. However, advancements in the identification of objective TBI biomarkers will aid in diagnostic precision while also improving treatment outcomes. This is especially important in a military setting, where objective TBI biomarkers can inform quick and efficient decision-making regarding a military personnel’s suitability to perform duties.

Advancements have been made towards understanding the impact of chronic low-level blast exposure on military personnel; however, gaps remain. While many studies have used eye-tracking technology to examine TBI, oculomotor assessment has not been studied within a military population. Eye-tracking technology provides a unique non-invasive and quantitative solution for examining the impact of low-level blast exposure on military members. Therefore, the purpose of this study was to determine whether long-term operators of low-level blast exposure or high-caliber weapons use displayed oculomotor behaviors that differed from controls.

## 2. Materials and Methods

The total number of participants was 25 (between 32–53 years; *M* = 40.8, *SD* = 5.76). All participants were male and members of elite Military units with lengths of service between 10–15 years (M = 13.7, SD = 1.2). The blast exposure group consisted of participants in the ‘Breachers Group’ (BC; *n* = 9) and ‘Gunners Group’ (GG; n = 9). Controls were in the ‘C Group’ (CG; *n* = 7).

Participants were excluded from the study if they met any of the following pre-screening conditions: neurological disorders (such as known concussion, Parkinson’s disease); vision-related issues that prevented the successful calibration of all 9 points (such as extreme tropias, phorias, static visual acuity greater than 20/400, cataracts, consumption of drugs or alcohol within 24 h of testing) [8,27,28,29].

Participants were selected for the blast exposure group if they had met either of the following conditions:Were trained specialty ‘Breachers’ within the Military Unit, where, by virtue of their job description, to gain entry to locations, they used low-level explosives. These operatives are therefore exposed to the risk of injury from debris, fragments or whole-body translation.Were trained specialty ‘Gunners’ within the Military Unit, where, by virtue of their job description, they were to operate high-caliber weapons.

Control group participants were Military personnel who were part of the elite units but were not Breachers or Gunners. All participants did not wear additional protective headgear beyond the standard helmet.

All participants provided informed consent to participate in this study in accordance with IRB procedure. All testing was conducted by vision specialists (e.g., optometrists, ophthalmologists) who had received and passed the RightEye training, education and protocol procedures prior to testing. Additionally, all data was collected within a morning session between 8–10 a.m.

Stimuli were presented via the RightEye tests on a Tobii I15 vision 15″ monitor fitted with a Tobii 90 Hz remote eye tracker and a Logitech (model Y-R0017) wireless keyboard and mouse. The participants were seated in a stationary (non-wheeled) chair that could not be adjusted in height. They sat in front of a desk in a quiet, private room. Participants’ heads were unconstrained. The accuracy of the Tobii eye tracker was 0.4° within the desired headbox of 32 cm × 21 cm at 56 cm from the screen. For standardization of testing, participants were asked to sit in front of the eye-tracking system at an exact measured distance of 56 cm, which is the ideal positioning within the headbox range of the eye tracker.

A Circular Smooth Pursuit (CSP) test was administered to all participants. Participants were asked to ‘follow the dot, on the screen, as accurately as possible with their eyes as it moved around in a circle’ (called Circular Smooth Pursuit (CSP)). The dot was 0.2 degrees in diameter and moved at a speed of 25 degrees of visual angle per second. The tests were taken with a black background with a white dot and lasted 20 s. The diameter of movement of the CSP circle was 20 degrees.

Two metrics were used to examine the eye movements while conducting the CSP test and included:

Fixation percentage: which accounts for the amount of time that the eye remains still when it should be following the target. It is measured in milliseconds.

Smooth pursuit variance: the average distance that the eye deviates from the ideal pathway. It is measured in millimeters.

The Sport Concussion Assessment Tool 2 was administered to all participants [30]. The SCAT is used by healthcare professionals, and is a standardized tool for the acute evaluation of suspected concussion [25]. The SCAT involves 22 questions asking the participant to rate ‘how they feel?’ on a scale of 0 (no symptom) to 1 or 2 (mild symptoms), 3 or 4 (moderate symptoms), and 5 or 6 (severe symptoms).

The nature of the study was explained to the participants, and all participants provided informed consent to participate. The study was conducted in accordance with the tenets of the Declaration of Helsinki. The study protocols were approved by the Institutional Review Board of East Carolina University. Following informed consent, participants were asked to complete a pre-screening questionnaire and an acuity vision screening where they were required to identify 4 shapes at a 4 mm diameter. If any of the pre-screening questions were answered positively or if any of the vision screening shapes were not correctly identified, the participant was excluded from the study.

Qualified participants who successfully passed the none-point calibration sequence completed the eye-tracking test. Written instructions on the screen and animations were provided before the test to demonstrate the appropriate testing behavior.

Once eye-tracking testing was complete, the participant completed the Sport Concussion Assessment Tool 2 [30].

## 3. Results

### 3.1. Data Analysis

The differences in the groups (Control vs. Blast) were analyzed on clinically verified data using JMP PRO 14.0 (SAS Institute; Cary, NC, USA). All variables were check for multicollinearity (Table 1). The comparison was evaluated using one-way univariate ANOVAs on the fixation stability measures, including: Visual Reaction Speed (RS), Fixation Percentage, Eye-Target Velocity Error, Smooth Pursuit Variance, and Saccadic Velocity. The alpha level was set at *p* < 0.05, and partial eta-squared (*η_p_*^2^) was used to determine the effect size. In addition, a series of receiver operating characteristic (ROC) curve analyses were plotted for the fixation stability variables. A significant area under the curve (AUC) with 95% confidence intervals (*p* < 0.05) was used to indicate the ability of each variable to differentiate concussed participants from non-concussed ones. A stepwise multivariable logistic regression model with an expert knowledge approach was used to assess the relationship between Control and Blast groups and circular smooth pursuit variables: Visual RS, Fixation Percentage, Eye-Target Velocity Error, Smooth Pursuit Variance, and Saccadic Velocity. The total number of variables were reduced to avoid collinearity and to include the variables with the most relevance to the research question. Global effect tests were used to determine if a predictor was significant at α = 0.05.

### 3.2. Sport Concussion Assessment Tool 2

Table 2 demonstrates a significant increase in SCAT−2 symptom severity and symptom scores (*p* = 0.05). For the blast-exposed participants, there was an 18% increase in balance problems; 20% increase in blurred vision; and approximately a 20% increase in headaches, nausea, and light sensitivity. Other increases were seen in neck pain, falling asleep, and eye strain. The remaining symptoms demonstrated little to no change pre and post exposure.

### 3.3. Fixation Stability Measures Analysis

The ANOVA results for the Visual Reaction Speed and Circular Smooth Pursuit Fixation Percentage demonstrated a significant main effect for Group [*F*(1, 24) = 5.336; *p* = 0.03, *η_p_*^2^ = 0.181] and [*F*(1, 24) = 10.313; *p* < 0.001, *η_p_*^2^ = 0.301], respectively (see Table 3). The ANOVA results for the Horizontal Smooth Pursuit Eye-Target Velocity error demonstrated a significant main effect [*F*(1, 24) = 6.951; *p* = 0.04, *η_p_*^2^ = 0.22] and Circular Smooth Pursuit: Smooth Pursuit Variance [*F*(1, 24) = 4.069; *p* = 0.049, *η_p_*^2^ = 0.144]. Further, the data demonstrated a significant effect for Vertical Saccades: Saccadic Velocity [*F*(1, 24) = 5.53; *p* = 0.027, *η_p_*^2^ = 0.187]; however, Targeting Displacement [*F*(1, 224) = 3.381; *p* = 0.067, *η_p_*^2^ = 0.293] demonstrated a non-significant difference between the Control and Blast groups.

### 3.4. Multivariable Logistic Regression

A stepwise multiple logistic regression analysis was conducted to evaluate how well the criterion variable blast status predicted the visual function. The predictors were the five smooth pursuit indices Visual RS, Fixation Percentage, Eye-Target Velocity Error, Smooth Pursuit Variance, and Saccadic Velocity, while the criterion variable was Blast status. The linear combination of Fixation Percentage and Saccadic Velocity was significantly related to the TBI status, χ^2^ = 14.109; *p* < 0.01, *R*^2^ = 0.459. The other predictors did not significantly contribute to the model and were removed (see Table 4). The final model accurately predicted 84.6% of the TBI status, with a sensitivity of 94% and specificity of 63%.

Among the smooth pursuit parameters, the ROC curves were significant for the Visual Reaction Speed, Fixation Percentage, and Smooth Pursuit Variance (see Table 5 and Figure 1). The remaining variables did not produce significant ROC curves and produced low AUC scores.

## 4. Discussion

The aim of this study was to determine whether long-term operators of low-level blast exposure or high-caliber weapons use displayed oculomotor behaviors that differ from controls. This was the first study to assess SPEM, saccades, and fixations within a military population exposed to repeated low-level blasts and high-caliber weapon use. The results revealed significant differences in SPEM, saccades, and fixations between the blast exposure group and control group. There was no significant difference between groups for targeting displacement. The results demonstrate that, in comparison to the control group, the blast exposure group’s eye movements were slower, stopped at more frequent points when following a target, traveled further from the target in both speed and direction, and showed higher rates of variation and inefficiency.

Oculomotor behavior has emerged as a sensitive biomarker for TBI, with an ability to differentiate the diagnosis severity [8]. The CSP is the ability to follow a target around in a circle while minimizing the amount of time that the eye remains still. Previous studies have observed that individuals with a TBI have longer fixation percentages when engaging in CSP [22]. The blast exposure group stopped moving their eyes significantly more often when compared to the controls. This dysfunction is implicated in frontal lobe planning and decision-making activities, only evident when a decision is required. These findings contribute to a clearer understanding of the impact that chronic low-level blast exposure has on the CSP fixation percentages of military personnel.

Variance in CSP is tracked in three segments of the pathway including middle, left/right, and up/down. Intact vestibulo-ocular reflexes require efficient functioning in a network of brain regions that spans the parietal and occipital lobes, premotor cortex, and brainstem. Inefficient, high-variance eye movements are often indicative of TBI in these regions [7]. The blast exposure groups had more than twice as much variation from the target than the control group when engaged in smooth pursuit activity. This data provides new insight into the relationship between repetitive low-level blast exposure and smooth pursuit variance in military populations.

The results revealed that the blast exposure group was significantly slower than the control group when engaged in saccadic eye movements. The slow saccadic eye movements suggest that the blast exposure group had difficulty rapidly and efficiently moving their eyes between targets. This is notable considering that multiple studies have observed a relationship between impaired saccadic eye movements and TBI [10,24]. These findings shed light on the potential for long-term low-level blast exposure to compromise the saccadic eye movements of military special operators.

Collectively, the significant differences in SPEM, fixations, and saccades observed in the blast exposure group are suggestive of TBI symptomology. This assertion is in line with prior research suggesting that compromised saccades, fixations, and smooth pursuits are implicated in TBI [9]. Unsurprisingly, poor oculomotor behavior in the blast exposure group was correlated with a higher symptom severity on the SCAT in comparison to the controls. Symptom severity was most evident through increased balance problems, blurred vision, headaches, nausea, and light sensitivity. The accuracy of the model was 84.6%, which suggests an excellent fit. Everything included, the data should be taken into account when considering the impact of long-term low-level blast exposure and high-caliber weapon use on the brain health of military personnel.

Practical implications converge on data that supports the refinement of military protective gear in addition to providing evidence of an objective TBI biomarker that can identify those with effects from being exposed to low-level blasts and higher caliber weapons training. Notably, our findings build on existing evidence that suggests neurological alterations may be implicated in chronic exposure to low-level blasts [2]. The military may benefit from using this data to inform interventions that better protect the neurological health of its service members.

Limitations of the current study include the fact that all 25 participants were male. However, this factor was unavoidable given a current restriction on women serving within the military units of the target population. A second limitation involves the CG’s exposure to low-level blasts. While the CG participants were not trained as Breachers or Gunners, most military personnel will have some exposure to blasts throughout their career, making it challenging to control for this variable. Additionally, factors such as smoking, medication use, and deployment history are unknown. It could be argued that the sample size is small; however, the employed model differentiated the blast exposure group from the control group and was substantiated by the overall statistical power of the test. Lastly, a potential limitation is the use of a stepwise regression, which has been criticized by some statisticians. A stepwise regression can be especially problematic with a large number of predictor variables; however, our variable selection was based originally on an expertise model and standard data reduction procedure. The final regression, albeit stepwise, does align with our previous work and is supported by the outcome of the ROC analysis.

## 5. Conclusions

This study was the first to examine the impact of long-term low-level blast exposure on the oculomotor behavior of military special operators. Future research should consider splitting the blast exposure group in order to discern whether oculomotor behavior differs between groups exposed to long-term low-level blasts and high-caliber weapon use. In conclusion, the results of the study found that the oculomotor behavior of military special operators exposed to long-term low-level blast exposure or high-caliber weapon use differed significantly from controls in (a) CSP fixation percentages, (b) CSP variance, and (c) saccadic eye movements, while poor oculomotor behavior was significantly correlated with a higher symptom severity on the SCAT.

## Figures and Tables

**Figure 1 brainsci-12-00679-f001:**
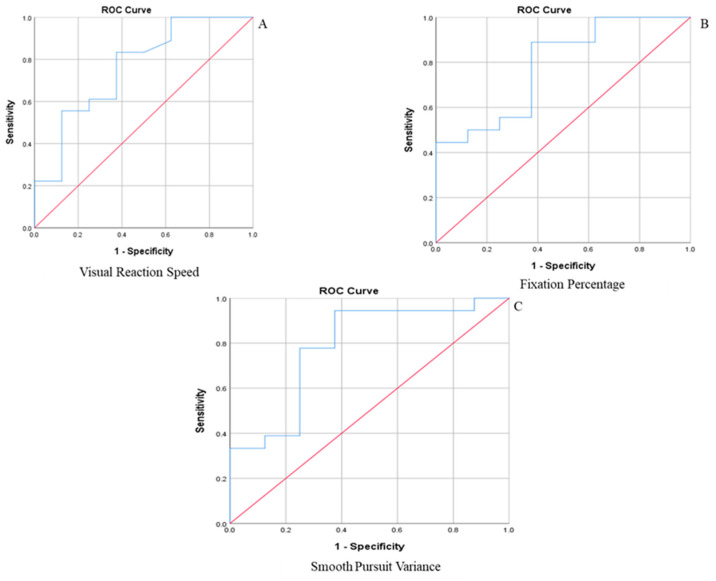
ROC for CSP’s: (**A**) Visual Reaction Time Speed (AUC = 0.706); (**B**) Fixation Percentage (AUC = 0.785); (**C**) Smooth Pursuit Variance (AUC = 0.785).

**Table 1 brainsci-12-00679-t001:** Strengths of the associations between the various eye-tracking parameters.

	1	2	3	4
1. Visual Reaction Speed				
2. Fixation Percentage	−0.14			
3. Smooth Pursuit Variance	0.32	0.46		
4. Eye-Target Velocity Error	−0.19	0.46	0.47	
5. Saccadic Velocity	0.65	−0.08	−0.53	−0.63

**Table 2 brainsci-12-00679-t002:** SCAT-2 symptom scores pre and post impact.

	Pre-SCAT-2 Symptom Severity Score	Post-SCAT-2 Symptom Severity Score
Control	0.40 (SD = 0.89)	0.63 (SD = 1.76)
Blast	20.36 (SD = 15.83)	32.87 (SD = 26.77)

**Table 3 brainsci-12-00679-t003:** Mean and standard deviation for fixation stability variables.

Group (n)	Visual RS	Fixation Percentage	Eye-Target Velocity Error	Smooth Pursuit Variance	Saccadic Velocity
Control	350.13 (51.96)	3.63 (0.721)	17.34 (1.62)	6.88 (3.95)	65.10 (13.96)
Blast Exposure	397.94 (47.31)	4.62 (0.726)	18.61 (0.863)	12.46 (7.306)	51.79 (11.97)

**Table 4 brainsci-12-00679-t004:** Estimated results for model coefficients.

		B	S.E.	Wald	df	Sig.	Exp(B)
Step 1	Fixation Percentage	2.091	0.933	5.021	1	0.025	8.09
	Constant	−7.786	3.759	4.291	1	0.038	0
Step 2	Fixation Percentage	3.161	1.519	4.33	1	0.037	23.594
	Saccadic Velocity	−0.169	0.098	2.989	1	0.084	0.844
	Constant	−2.385	4.658	0.262	1	0.609	0.092

Beta coefficient (B); Standard error (SE); Wald chi-squared test (Wald); Degrees of freedom (df); Statistical significance (Sig); Odds ratio (Exp(B)).

**Table 5 brainsci-12-00679-t005:** Summarization of outcomes for the ROC curve analysis.

Variable	AUC	S.E.	*p*
Visual Reaction Speed	0.706	0.107	0.037
Fixation Percentage	0.785	0.098	0.023
Smooth Pursuit Variance	0.785	0.104	0.023

Area under the curve (AUC); Standard error (S.E.); Probability value (*p*).

## Data Availability

The authors will make the data available should someone request it.

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
