# Peer review of "Long-Term Effects of Low-Level Blast Exposure and High-Caliber Weapons Use in Military Special Operators"

_brainsci, 2022, doi:10.3390/brainsci12050679_

Round 1

Reviewer 1 Report

This manuscript describes an interesting study that examined the association between different military specialties associated with exposure to blast and high caliber weapon use and smooth pursuit eye-tracking. The authors hypothesize that the eye-tracking task may provide an index blast-related traumatic brain injury and results were generally supportive of this. Unfortunately, the submission suffered from several serious weaknesses.

The introduction seemed to overstate the novelty and innovation of using eye-tracking to assess TBI. A cursory search of the literature shows a few dozen empirical studies on this topic including this meta-analysis of 19 such studies: Revathy Mani, Lisa Asper & Sieu K Khuu (2018) Deficits in saccades and smooth-pursuit eye movements in adults with traumatic brain injury: a systematic review and meta-analysis, Brain Injury, 32:11, 1315-1336, DOI: 10.1080/02699052.2018.1483030

The use of stepwise regression was problematic.  This approach has been widely criticized and has fallen out of favor among statisticians for various reasons including its tendency to capitalizing on chance associations in the dataset and poor replicability.  It would be more defensible to regress group on all of the IVs simultaneously. 

The analyses and results should include a correlation matrix listing the strength of the bivariate associations between the various eye-tracking parameters.

There was confusing information presented about the Sport Concussion Assessment tool (Table 1) which was apparently administered pre- and post-impact, but I found no mention of what the impact was?  The concussion scores apparently increased from pre- to post, but from pre to post what?  Perhaps this is a revised version of a manuscript that previously had an experimental manipulation?  It seemed odd that the blast exposed groups were repeatedly referred to as the “experimental” when there was no experiment described.

The Sport Concussion Assessment tool needs a more elaborate description and information about its psychometric properties, reliability, and validity. 

There was no attention given to the demographic and other relevant characteristics of the different groups which raises concern that findings may have been confounded by important but unmeasured third variables (e.g., age, deployment history, length of service, time of day, smoking, medication use, etc).  

Reviewer 2 Report

This is a very good study; however, the following points needs to be addressed.

  1. Total number of participants were 25 not 26 based on 9+9+7 which is very low to draw conclusions
  2. No details were provided reading the subjects chosen such as how many hours they expose to sound per day or week. Also, using any protective gear to avoid such noise. Including those details will make the study more effective.
